# The Effect of Long-Term Ageing at 475 °C on Microstructure and Properties of a Precipitation Hardening MartensiticStainless Steel

Vlastimil Vodárek [1,*], Gabriela Rožnovská [2], Zdeněk Kuboň [2], Anastasia Volodarskaja [1]
and Renáta Palupčíková [1]

[1] Department of Materials Engineering and Recycling, Faculty of Materials Science and Technology, VSB-Technical University of Ostrava, 708 30 Ostrava, Czech Republic
[2] Material and Metallurgical Research, Ltd., Pohraniční 31, 706 02 Ostrava, Czech Republic
[*] Correspondence: vlastimil.vodarek@vsb.cz

**Abstract:** The effect of long-term ageing (1000, 2000, and 3000 h) at 475 °C on mechanical properties, microstructure, and substructure of CUSTOM 465® maraging stainless steel was studied. The additional precipitation of nanometric particles of η-Ni₃Ti phase in partly recovered lath martensite and decomposition of the BCC solid solution accompanied by the formation of nanometric Cr-rich $\alpha'$ particles were identified. The fraction of reverted austenite in the final microstructure gradually increased with time of ageing at 475 °C. Ageing resulted in a gradual slight decline (up to 10%) in yield strength, ultimate tensile strength, and hardness. On the other hand, for all ageing, dwells ductility and impact energy values remained almost unchanged. The reason for this phenomenon lies in the gradual increase in the fraction of reverted austenite during long-term ageing at 475 °C and at the same time in the sluggish kinetics of microstructural changes in lath martensite. No susceptibility to 475 °C embrittlement was proved.

**Keywords:** CUSTOM 465® stainless steel; mechanical properties; precipitation processes; η-Ni₃Ti phase; Cr-rich $\alpha'$ phase; reverted austenite; twinning in martensite; long-term ageing

## 1. Introduction

Maraging steels represent an important group of ultrahigh-strength steels (a yield strength of more than 1380 MPa) where an excellent combination of high strength and good toughness is achieved by optimized chemical composition of steels and heat treatment procedure consisting of quenching and ageing [1,2]. Ageing of soft, low-carbon lath martensite results in significant strengthening caused by intensive precipitation of nanometric intermetallic phases and it is accompanied by stabilization of some reverted austenite, which is beneficial to ductility and toughness [3]. Nickel in maraging steels plays an important role in the stabilization of reverted austenite during ageing below the $A_{c1}$ temperature [2].

CUSTOM 465® is a martensitic, precipitation-strengthened stainless steel used in demanding applications where a tensile strength of more than 1380 MPa is required [4]. Originally, it was developed for the main landing gears of large aircrafts, but it is also used for high-pressure pump components and drilling rigs in oil and gas extraction under extreme conditions, on steam turbine parts, and in surgical instruments [1]. A unique combination of high strength, toughness, fatigue strength, and corrosion resistance can be obtained after ageing of low-carbon lath martensite in the temperature range from 480 to 593 °C [4–8]. The peak-aged condition can be achieved after ageing around 520 °C [9]. Higher ageing temperature increases the toughness, but at the same time lowers the strength. CUSTOM 465® stainless steel is based on 11 wt.% Ni and 12 wt.% Cr and is also alloyed with titanium and molybdenum [4].

A lot of attention has been devoted to investigations into precipitation reactions in maraging steels and their effects on properties [3]. Advanced experimental techniques, such as high-resolution transmission electron microscopy (HRTEM) and atom probe tomography (APT), are used today for better understanding of segregation and precipitation phenomena which govern properties of these steels [10–12]. In Cr–Ni–Ti–(Mo) maraging steels, rod-like $\eta$-Ni$_3$Ti particles are regarded as the most important minor phase, and are responsible for intensive precipitation hardening of lath martensite [9,13–19]. Chen et al. [9] deduced from the patent analysis that high tensile strength levels can be achieved in Cr–Ni–Ti–(Mo) precipitation-hardened steels when the Ni:Ti atomic ratio is close to 3. This value corresponds to the composition stoichiometry of the $\eta$-Ni$_3$Ti phase. APT results indicate that molybdenum partitions into the $\eta$-Ni$_3$Ti precipitate core and substantially increases the number density of these particles [11–13]. The thermodynamic calculations indicate that molybdenum not only enhances the chemical driving force, but also reduces the strain energy for the $\eta$-Ni$_3$Ti nucleation. At longer times of aging, molybdenum atoms are rejected to the $\eta$-Ni$_3$Ti/matrix interface and this can be accompanied by precipitation of molybdenum-enriched particles on the outer surface of $\eta$-Ni$_3$Ti particles [12]. According to Thuvander et al. [11], these effects of molybdenum in Cr–Ni–Ti–Mo maraging steels result in a faster ageing response at short times, but in an increased resistance to over-ageing at longer times. Controversial opinions exist on the nucleation of $\eta$-Ni$_3$Ti particles. Some researchers reported that this phase directly heterogeneously nucleates on dislocations [15]. However, APT studies of Thuvander et al. [10] indicate that firstly, clusters of nickel and titanium are formed and they are subsequently transformed to nuclei of the $\eta$-Ni$_3$Ti phase. Thermodynamic data prove that this minor phase is stable in the BCC matrix up to about 600 °C [18]. Dimensional stability of $\eta$-Ni$_3$Ti particles rapidly deteriorates at temperatures above 540 °C [9]. It is a hexagonal close-packed phase with lattice parameters of a = 0.255 nm and c = 0.831 nm [6]. It obeys a Burgers orientation relationship with the martensitic matrix, i.e., $\{011\}_\alpha//\{0001\}_\eta$ and $<11\bar{1}>_\alpha//<11\bar{2}0>_\eta$ [14,15]. Precipitation introduces elastic stresses and strain into lath martensite, and thus it increases strengthening of alloys. Calculations of elastic fields and interaction between $\eta$-Ni$_3$Ti precipitates in lath martensite of CUSTOM 465® steel predict that $\eta$-Ni$_3$Ti precipitates will favor an ellipsoidal morphology with a high aspect ratio [20]. Experimentally, needle-shaped precipitates of b/a = 3 were observed in the peak-aged condition. The interaction energy of two precipitates reaches up to 7% of the self-energy [20]. Intensive precipitation of $\eta$-Ni$_3$Ti particles takes place simultaneously with the formation of reverted austenite. Thermodynamic calculations indicate that dissolution of $\eta$-Ni$_3$Ti precipitates does not control the austenite reversion, although both phases compete for nickel [9].

In maraging steels, partial reversion of martensite to austenite can occur at temperatures of ageing deep below the equilibrium $A_{c1}$ temperature [3]. The fraction of reverted austenite in the final microstructure is controlled by chemical composition of steels, the ageing temperature, and time of ageing [21–25]. Due to differences in the thermal expansion coefficients of austenite and ferrite, elastic stresses or dislocations can form around reverted austenite/martensite interfaces [9]. The important role in chemical stabilization of reverted austenite belongs to redistribution of nickel and carbon. Morphology of reverted austenite islands depends on the alloy composition and the applied heat treatment. Globular (blocky) particles of reverted austenite preferentially nucleate on prior austenite grain boundaries, while thin films of reverted austenite usually grow along martensite lath boundaries [3]. Reverted austenite declines strength, but it is beneficial for increasing ductility and toughness of maraging steels [9]. Reverted austenite is free of precipitates causing precipitation hardening. Thermal stability of reverted austenite is very good. On the other hand, deformation can cause deformation-induced transformation of reverted austenite to martensite (TRIP effect), especially in coarse globular particles of reverted austenite. Resistance of reverted austenite particles against martensitic transformation is affected by their chemical composition, morphology, and size [26,27].

Controversial results exist about the possibility of $\omega$ phase precipitation in maraging steels. The metastable $\omega$ phase was firstly reported in a precipitation-hardened Fe–Ni–Co–Mo alloy by Yedneral et al. [28]. It is a hexagonal phase with lattice parameters of $a_\omega = \sqrt{2}a_{BCC}$ and $c_\omega = \sqrt{\frac{3}{2}}a_{BCC}$. Detailed characterization of precipitation reactions in Cr–Ni–Ti–Mo maraging steels revealed the following precipitation sequence during aging at 500 °C [12]: solid solution $\rightarrow$ $\eta$-Ni$_3$Ti $\rightarrow$ $\eta$-Ni$_3$Ti + Cr-rich $\alpha'$ + $\omega$. Diffraction patterns recorded on Mo-enriched particles corresponded to the A$_7$B$_2$-type $\omega$ phase and also the atomic arrangement of the precipitate matched the simulated structure of $\omega$ phase [12]. However, experimental findings indicate that composed diffraction patterns of this phase and the martensitic matrix are very similar to composed diffraction patterns of twinned martensite [29,30]. Casillas et al. [31] studied the composed diffraction patterns in high-carbon steels, where transformation twins exist in plate (lenticular) martensite [8,32]. They found that extra spots in the zone axis $[011]\alpha$ can be indexed either as $\omega$ phase or {112} <111> twins in martensite. Experimental and simulation results in high-carbon steels confirmed that transformation twins were the right solution [31]. Double diffraction spots in composed diffraction patterns originated from overlapped twin-matrix or multiple-twin regions. These double diffraction spots in composed diffraction patterns in the $[011]\alpha$ zone axis form due to the unique crystallographic relationship between the matrix and twins [33].

Some maraging steels contain an addition of copper [34]. Thermo-Calc calculations show that the solubility of Cu in a martensitic matrix with low carbon content decreases with increasing molybdenum content [11]. Cu-rich precipitates in the BCC matrix can undergo gradual changes. Firstly, nucleation of BCC Cu-rich precipitates was reported in the BCC matrix. Later it was discovered that these BCC precipitates transform at a critical diameter into semicoherent 9R twinned close-packed structures, followed by relaxation into a 3R structure, and finally they transform to the FCC unit cell [35,36]. It has been assumed that FCC Cu-rich precipitates act as nucleation sites for precipitation of reverted austenite because they have identical structures and similar lattice parameters [3].

Long-term thermal exposure of Fe–Cr-based alloys containing from 12 to 70 wt.% Cr between 425 and 550 °C may significantly deteriorate properties of stainless steels [37]. This 475 °C embrittlement manifests itself by increased hardness and ductile–brittle transition temperature [38]. The reason for this phenomenon is the existence of the miscibility gap in the binary Fe–Cr phase diagram [37]. At a temperature of below 550 °C, the BCC solid solution $\alpha$ decomposes into fine areas rich in chromium ($\alpha'$ phase) and chromium-depleted areas [39]. The Cr-rich $\alpha'$ phase has a BCC crystal structure and contains from 61 to 83 wt.% Cr [3]. There are two basic mechanisms of $\alpha$ solid solution decomposition: either spinodal decomposition or nucleation and the growth of $\alpha'$ phase particles [40]. The rate and extent of 475 °C embrittlement is a function of chromium content; at least 100 h of thermal exposure is required for embrittlement of low- and medium-chromium steels. High-chromium alloys may exhibit loss of ductility and toughness at much shorter times [37]. Miller et al. [41] reported that the addition of nickel to Fe–Cr-based alloys raises the critical temperature of the miscibility gap. The Cr-rich $\alpha'$ phase in Ni–Cr–Ti–Mo maraging steels is also affected by the precipitation of the $\eta$-Ni$_3$Ti phase due to the change in the phase stability of the matrix. A decline of the nickel content in the matrix associated with $\eta$-Ni$_3$Ti precipitation enhances the stability of the matrix and inhibits the decomposition of the matrix in Ni–Cr–Ti–Mo maraging steels. As a result, the volume fraction of Cr-rich precipitates in Ni–Cr–Ti–Mo maraging steels is significantly lower than that in the Ni–Cr–Mo steels [12].

A lot of effort has been devoted to investigations into microstructure–property relationships in CUSTOM 465$^{\circledR}$ alloy. The attention has been paid not only to optimal parameters of ageing during quality heat treatment but also to long-term stability of properties and microstructure at elevated temperatures. The published data [4] show the effect of thermal exposure in the temperature interval of 316–482 °C for 1000 h on mechanical properties of the alloy. Long-term exposure at 371 and 427 °C was accompanied by a slight strengthening and a gradual decrease in impact toughness. On the other hand, long-term exposure at temperature of 482 °C indicated a decline in strength values while impact energy increased [4].

CUSTOM 465® stainless steel has a chromium content close to the lower limit which is regarded to be critical from the point of view of the Fe–Cr solid solution decomposition and developing of 475 °C embrittlement during long-term exposure at temperatures in the range of about 425 to 550 °C.

The main objective of this paper is to investigate the effects of exposure at 475 °C for 1000, 2000, and 3000 h on the microstructural evolution and mechanical properties of CUSTOM 465® stainless steel. Results are expected to deepen the knowledge about long-term microstructural stability, precipitation reactions, and mechanical properties of Ni–Cr–Ti–Mo maraging steels and their susceptibility to 475 °C embrittlement.

## 2. Experimental Material and Procedures

Investigations were carried out on a forged rod with a diameter of $\phi$ 140 mm and a length of 500 mm made of CUSTOM 465® stainless steel. Chemical composition of the rod is shown in Table 1.

**Table 1.** Chemical composition of the cast investigated (wt.%).

| C | S | P | Mn | Si | Ni | Cr | Mo | Ti | Al | V | W | Cu |
|------|-------|-------|------|-------|-------|-------|------|------|-------|--------|-------|------|
| 0.01 | 0.001 | 0.006 | 0.01 | 0.048 | 10.82 | 11.07 | 0.93 | 1.55 | 0.054 | <0.003 | 0.006 | 0.57 |

Quality heat treatment and long-term ageing were carried out in an electric furnace with a protective atmosphere. Transformation temperatures during heating and cooling of the alloy were determined using the quenching dilatometer Bähr DIL 805A. The heating rate in the interval of phase transformations was 1 °C/min. The cooling rate from the temperature of 950 °C was 3 °C/min up to a temperature of 35 °C. Temperature of the end of martensitic transformation was lower than the temperature at the end of the measurement and therefore could not be determined. Microstructure and basic mechanical properties were studied in the state after quality heat treatment and after additional ageing at 475 °C for 1000, 2000, and 3000 h. Microstructural characterization was carried out using light microscopy (LM), scanning electron microscopy (SEM), transmission electron microscopy (TEM), and X-ray diffractometry (XRD). XRD measurements were carried out using a Bruker—AXS D8 Advance diffractometer (Bruker, Karlsruhe, Germany) equipped with a position sensitive detector LynxEye. The following parameters were applied: radiation $CuK_\alpha$, absorption filter Ni, voltage 40 kV, current 40 mA, step 0.014° in the interval of 2θ angles 40–135°, and digital processing of data using software Bruker Diffract Suite. Diffraction database PDF-2 was applied for qualitative analysis; quantification was based on the Rietveld's method using the software Bruker Topas, version 4.2.

Metallographic samples were oriented perpendicular and parallel to the rod's axis and were cut in approximately one-quarter of the diameter of the rod. Polished samples were etched in a V2A solution (a mixture of 10 mL $HNO_3$, 100 mL HCl, and 100 mL water) and observed in an Olympus GX51 microscope (Olympus Corporation, Tokyo, Japan). SEM was performed using a Quanta 450FEG microscope (ThermoFisher Scientific, Brno, Czech Republic) equipped with X-ray microanalysis (EDX) and electron backscattered diffraction (EBSD) facilities. Topography of fracture surfaces was recorded using a signal in secondary electrons (SEs). EBSD mapping was carried out on specimens after polishing on colloidal silica at accelerating voltage of 15 kV, and a step size of 0.2 μm was applied. TEM analysis was performed on a JEM 2100 microscope (JEOL Ltd., Tokyo, Japan) equipped with an EDX analyzer. Identification of phases was carried out by a combination of EDX microanalysis and electron diffraction. Both carbon-extraction replicas and thin foils were used. Thin foils were prepared by twin-jet electrolytic polishing to perforation using a Struers Tenupol 2 equipment (Struers, Copenhagen, Denmark). Electropolishing was carried out in an electrolyte consisting of 5 vol.% perchloric acid, 20 vol.% glycerol, and 75 vol.% ethanol at −5 °C and a potential of 20 V. Tensile tests were carried out at room temperature according to the ASTM E8 standard on an MTS 810 testing machine (MTS Systems Corporation, Eden Prairie, MN, USA) at a strain rate of $1 \times 10^{-3}$ s$^{-1}$. Tensile bars of $d_o$ = 6 mm and l = 5$d_o$

were oriented in the tangential direction of the original rod. Impact tests were carried out on samples oriented in the same way, with a V-notch and dimensions of $5 \times 10 \times 55$ mm. Tests were performed at room temperature on a Charpy instrument 300 J according to the ASTM E23 standard. Hardness measurements were carried out on metallographic specimens using the Vickers method at a load of 30 kg.

Transformation temperatures, as determined by analyses of heating and cooling curves, are summarized in Table 2.

**Table 2.** Transformation temperatures of the cast investigated.

| $A_{c1}$ (°C) | $A_{c3}$ (°C) | $M_s$ (°C) |
|:---:|:---:|:---:|
| 589 | 738 | 131 |

During quality heat treatment, the rod was quenched from an austenitizing temperature of 862 °C into oil and immediately chilled at $-80$ °C. XRD measurements proved that the fraction of retained austenite in lath martensite was less than 2 wt.%. Subsequent age hardening was carried out at a temperature of 524 °C for 8 h. The ageing parameters were designed to be close to the peak-aged condition of the steel investigated [9]. Results of a tensile test, hardness HV 30 evaluation, and a Charpy test at room temperature carried out on the rod after quality heat treatment are stated in Table 3.

**Table 3.** Mechanical properties after quality heat treatment.

| $R_{p0.2}$ (MPa) | $R_m$ (MPa) | L (%) | R.A. (%) | HV 30 | KV (J) |
|:---:|:---:|:---:|:---:|:---:|:---:|
| 1531 | 1617 | 11.3 | 44.0 | 500 | 18 |

L: elongation; R.A.: reduction in area.

## 3. Results and Discussion

Microstructure of the rod after quality heat treatment consisted of aged lath martensite (Figure 1a). The prior austenite grain size was evaluated using the EN ISO 643 standard as G = 6. Most nonmetallic inclusions in the matrix were determined as TiX particles, where X = N or C. These sharp-edged particles formed during solidification, and sometimes heterogeneous nucleation on alumina particles was observed. Ageing resulted in enrichment of these particles in molybdenum (up to 10 at.%).

The typical size of TiX particles was several micrometers. That is why they were not very effective at pinning of austenite grain boundaries during austenitizing. Figure 1b documents segregation bands enriched in nickel and a string of TiX particles parallel to the rod's axis and some defects at TiX/matrix interfaces. Figure 1c shows detail of aged lath martensite and a TiX inclusion.

The fraction of austenite, as determined by XRD technique, was $6.8 \pm 0.3$ wt.%. It means that ageing at 524 °C was accompanied by the formation of about 5 wt.% of reverted austenite. Globular particles of reverted austenite preferentially formed in segregation bands which were enriched with nickel. Figure 2a,b show an IPF map of lath martensite and a phase map, where heterogeneous distribution of globular particles of reverted austenite can be seen. It is worth noting that resolution of these EBSD images does not allow to visualize thin films of reverted austenite along martensite lath boundaries.

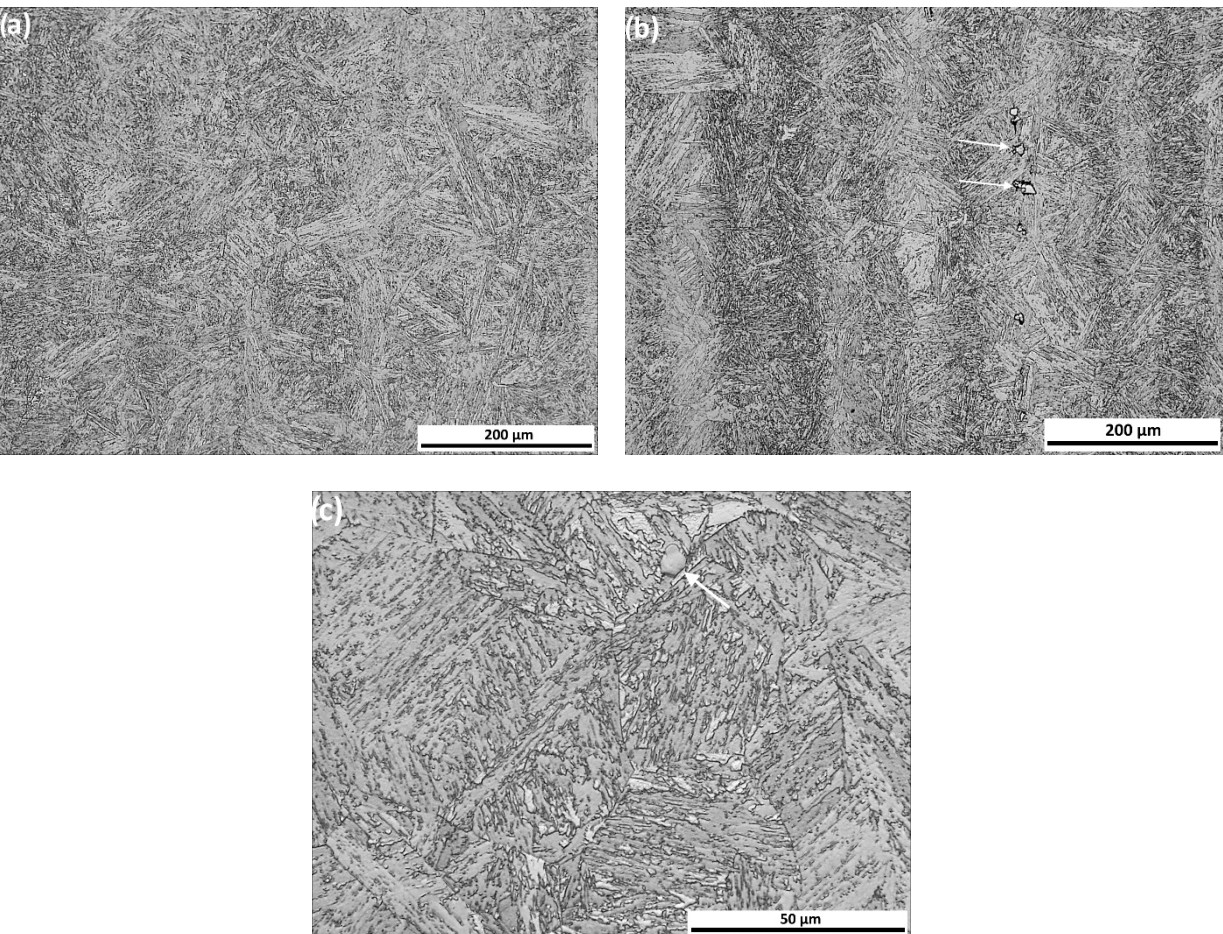

**Figure 1.** Microstructure after quality heat treatment, longitudinal section, (**a**) aged lath martensite, (**b**) seg-regation bands in aged lath martensite and a string of TiX inclusions (arrows), (**c**) a detail of aged lath martensite, an arrow marks a TiX particle, LM.

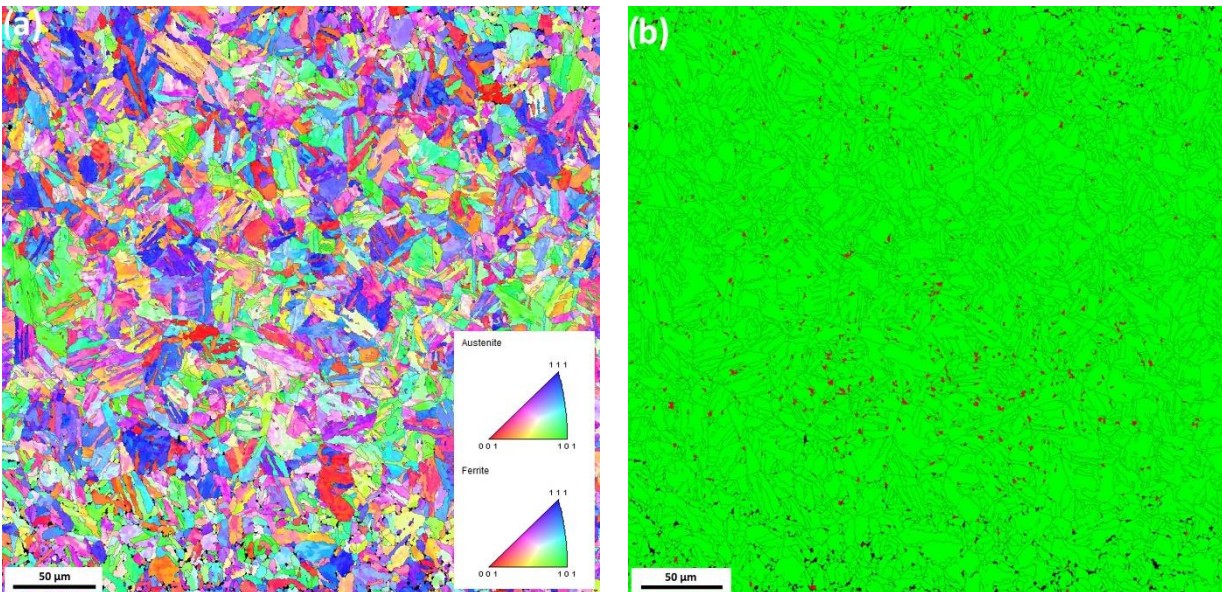

**Figure 2.** (**a**) IPF orientation map, aged lath martensite after quality heat treatment, (**b**) phase map; green = martensite, red = reverted austenite.

Ageing at 524 °C was accompanied by intensive precipitation of fine rod-like particles in martensitic laths (Figure 3a,b). These particles were identified as the η-Ni$_3$Ti phase. The average length of these particles, as determined by image analysis, was 20.1 ± 7.1 nm. Furthermore, a low number density of globular particles rich in molybdenum, titanium, chromium, and iron was detected on prior austenite grain and martensitic lath boundaries. The effect of these medium-sized particles on precipitation strengthening was negligible. Intensive precipitation of rod-like η-Ni$_3$Ti particles in lath martensite during ageing caused an increase in martensite hardness by approximately 75%. Reverted austenite particles were free of precipitates (Figure 3b).

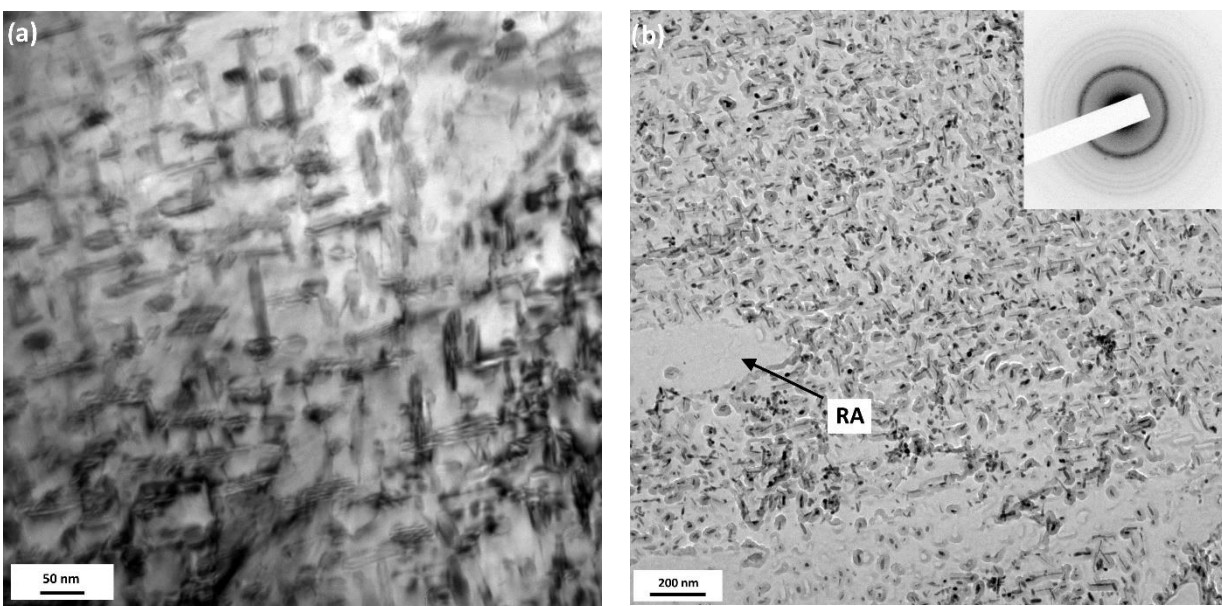

**Figure 3.** Intensive precipitation of η-Ni$_3$Ti rod-like particles in aged lath martensite after quality heat treatment; (**a**) thin foil, (**b**) carbon extraction replica. Inset: ring diffraction pattern of the η-Ni$_3$Ti phase; RA = reverted austenite free of precipitates, TEM.

Results of tensile tests and hardness HV 30 measurements on samples after long-term ageing at 475 °C for periods of 1000, 2000, and 3000 h are summarized in Table 4. Trends of mechanical property changes after long-term ageing at 475 °C are evident from graphical representation of average values shown in Figures 4 and 5. The results show a gradual smooth decline of yield strength and ultimate tensile strength with the ageing dwell. The total decrease of strength values after 3000 h exposure was only 10% in yield strength and 8% in case of ultimate tensile strength. The R$_p$0.2/R$_m$ ratio gradually declined with prolonging the ageing time. Results of HV 30 testing show the identical trend. Both elongation and reduction in area values slightly increased after 1000 h exposure. Prolonging of the ageing time to 2000 and 3000 h did not cause any changes in ductility.

**Table 4.** Tensile properties after long-term ageing at 475 °C.

| Sample | Dwell (hours) | R$_p$0.2 (MPa) | R$_m$ (MPa) | R$_p$0.2/R$_m$ | L (%) | R.A. (%) | HV 30 |
|---|---|---|---|---|---|---|---|
| C 1/1 | 1000 | 1461 | 1550 | 0.942 | 14.0 | 56.7 | - |
| C 1/2 | | 1463 | 1555 | 0.941 | 14.0 | 56.7 | 494 |
| C 2/1 | 2000 | 1435 | 1528 | 0.939 | 14.0 | 56.7 | - |
| C 2/2 | | 1435 | 1530 | 0.938 | 14.0 | 56.7 | 472 |
| C 3/1 | 3000 | 1376 | 1488 | 0.924 | 13.3 | 54.4 | - |
| C 3/2 | | 1369 | 1480 | 0.925 | 15.0 | 57.9 | 460 |

L: elongation; R.A.: reduction in area.

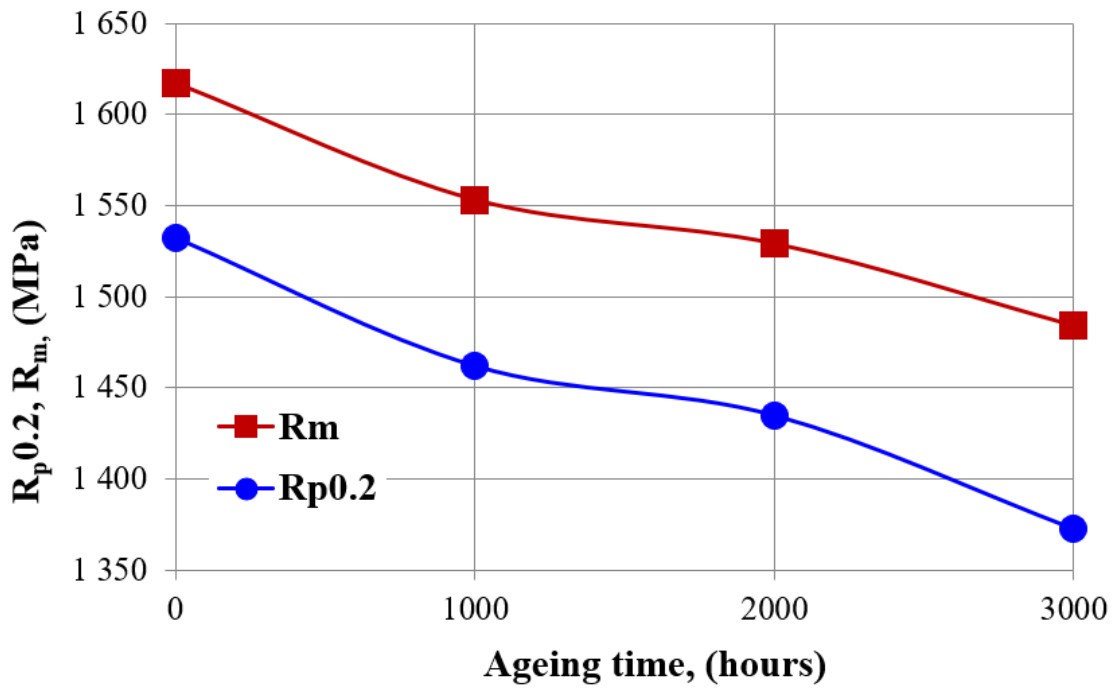

**Figure 4.** Changes of yield stress (R$_p$0.2) and ultimate tensile strength (R$_m$) during long-term ageing at 475 °C.

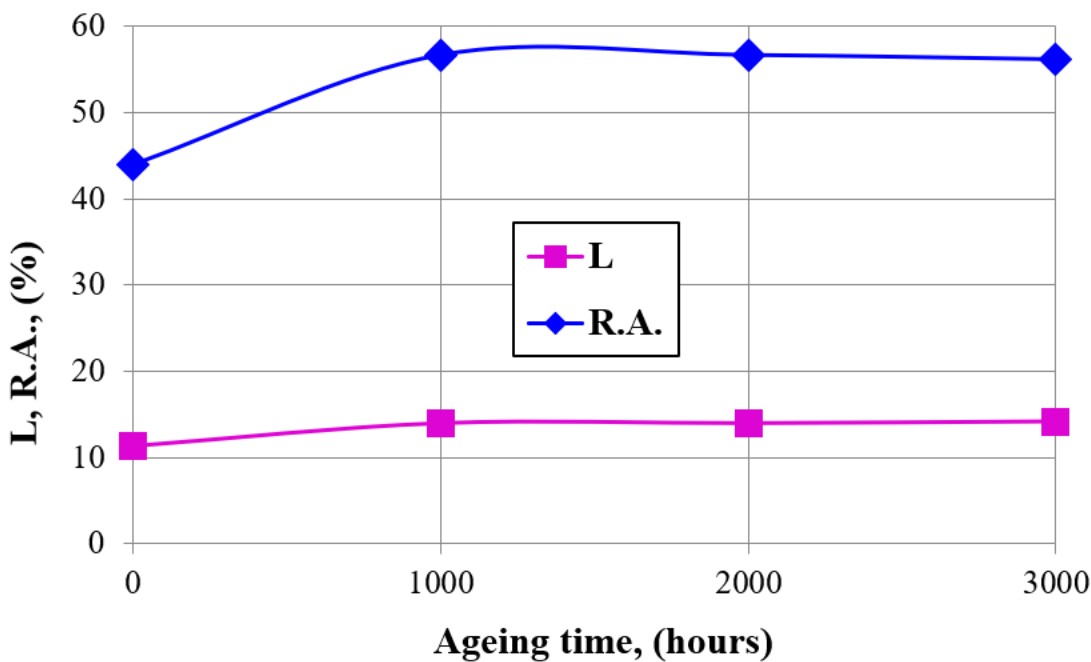

**Figure 5.** Changes of elongation (L) and reduction of area (R.A.) during long-term ageing at 475 °C.

Results of impact energy evaluation after long-term ageing at 475 °C are shown in Table 5. It is evident that ageing at 475 °C up to 3000 h had almost no effect on impact energy values at room temperature.

**Table 5.** Changes of impact energy values after long-term ageing at 475 °C.

| Sample | Dwell (hours) | KV (J) |
|---|---|---|
| C 1/1 | | 16 |
| C 1/2 | 1000 | 15 |
| C 2/1 | | 16 |
| C 2/2 | 2000 | 17 |
| C 3/1 | | 18 |
| C 3/2 | 3000 | 18 |

A visual inspection of the broken tensile bars revealed that the appearance of the fracture surface was tough with significant shear tearing along the perimeter of testing bars. Fractographic analysis proved that fracture surfaces consisted exclusively of the transcrystalline dimple fracture (Figure 6a,b). Sharp-edged particles, which were usually identified as TiX inclusions, were often found in dimples. These particles acted as stress concentrators. No significant differences in fracture surfaces were noted between samples after quality heat treatment and after long-term ageing at 475 °C.

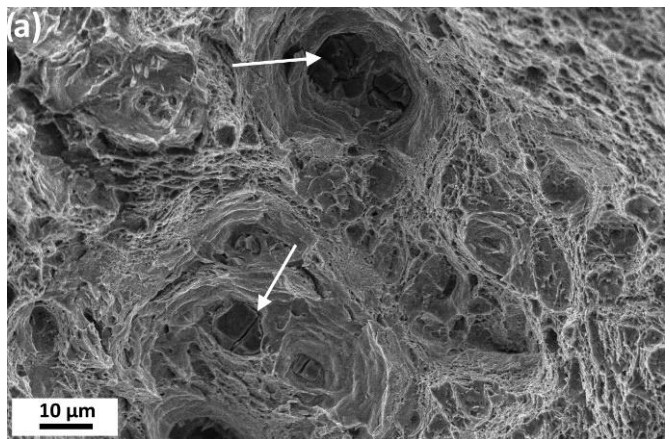 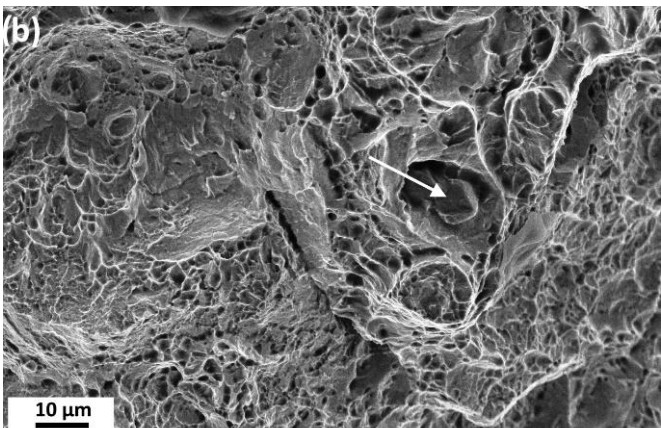

**Figure 6.** Ductile dimple fracture surfaces of tensile samples tested at room temperature: (**a**) after quality heat treatment; (**b**) after ageing at 475 °C for 3000 h. Arrows show sharp-edged TiX particles in some dimples.

Figure 7a shows the topography of the fracture surface of the Charpy sample in the state after quality heat treatment. It corresponds to a mixture of transgranular cleavage facets and dimple ductile fracture [42]. The fraction of brittle fracture is significantly greater. This mixed mode of fracture reflects the fact that microstructure consists of ductile reverted austenite and aged martensite strengthened by heavy precipitation. Long-term annealing of CUSTOM 465® stainless steel at 475 °C did not affect the fracture mode. Figure 7b shows the topography of the fracture surface of the Charpy sample after ageing at 475 °C for 3000 h. The fracture surface consists of a mixture of transgranular cleavage facets and dimple ductile fracture. Again, the fraction of brittle fracture is significantly greater.

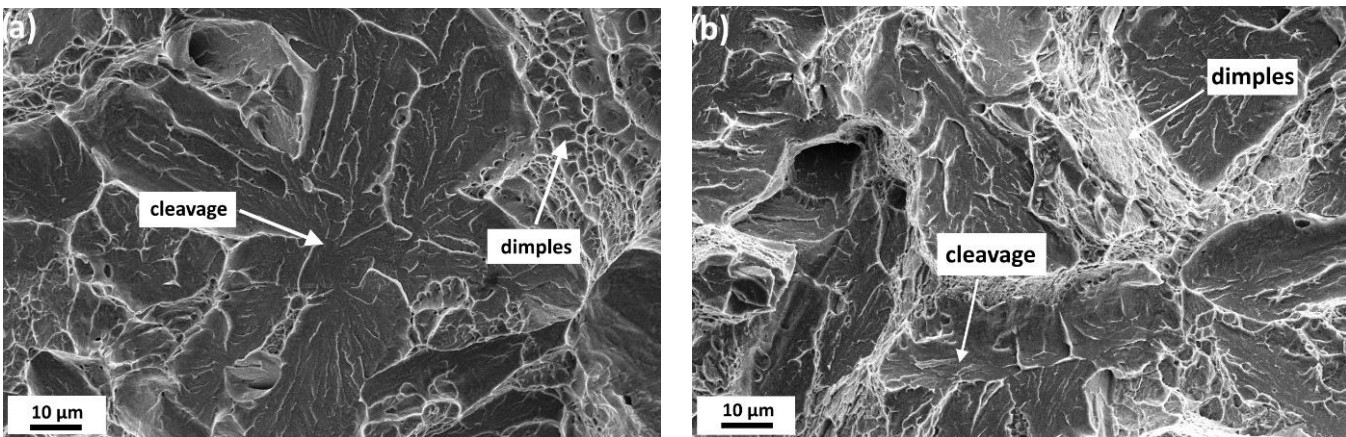

**Figure 7.** Topography of the fracture surfaces of Charpy samples formed by a mixture of transcrystalline cleavage and dimple ductile fracture: (**a**) after quality heat treatment; (**b**) after ageing at 475 °C for 3000 h.

Long-term exposure at 475 °C was accompanied by additional precipitation of η-Ni$_3$Ti particles. The driving force for this additional precipitation was given by supersaturation of the martensitic matrix in nickel and titanium. Concentration of these elements in lath martensite after ageing at 524 °C was higher than their equilibrium concentration at 475 °C [40]. Figure 8a,b show the typical precipitation of rod-like η-Ni$_3$Ti particles in lath martensite after 3000 h exposure at 475 °C. Ageing was accompanied by gradual recovery of martensite and slow growth of η-Ni$_3$Ti precipitates.

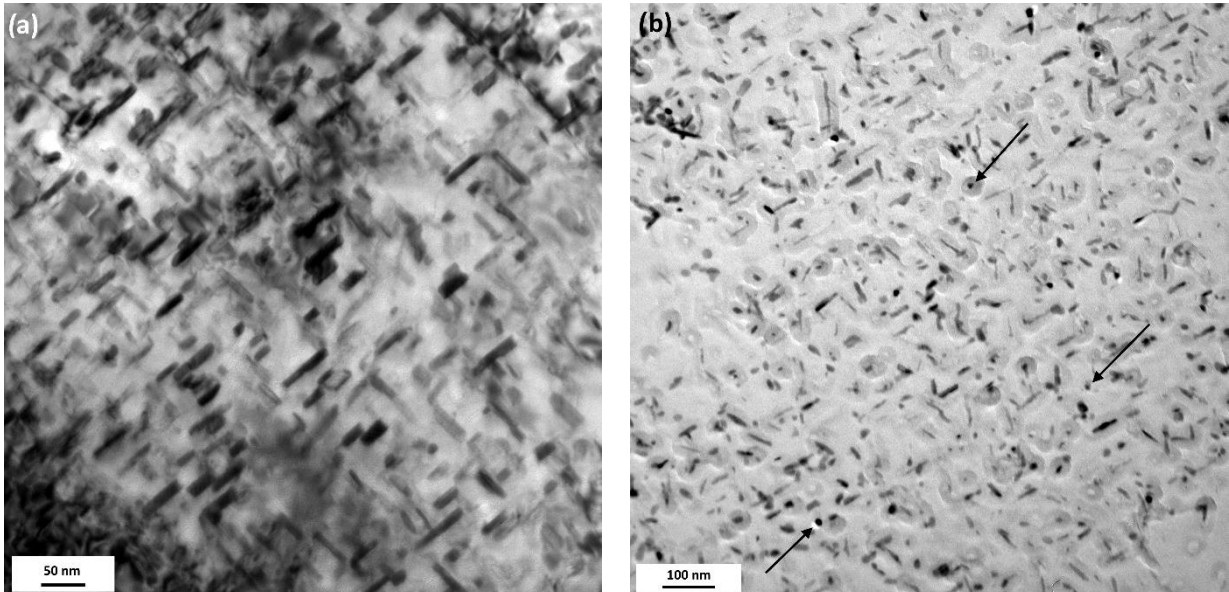

**Figure 8.** Precipitation in lath martensite after ageing at 475 °C for 3000 h: (**a**) thin foil; (**b**) carbon extraction replica. Arrows mark Cr-rich α′ phase particles.

Figure 9a,b document the variation of the length of η-Ni$_3$Ti rods in the state after quality heat treatment and after ageing at 475 °C for 3000 h, respectively. The average length of η-Ni$_3$Ti rod-like particles after the longest ageing dwell was determined as 30.5 ± 12.4 nm.

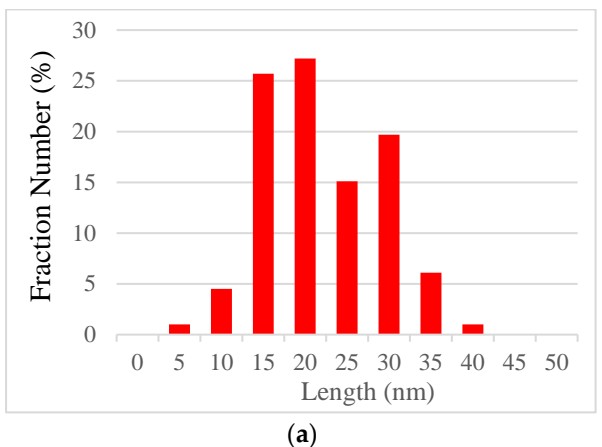
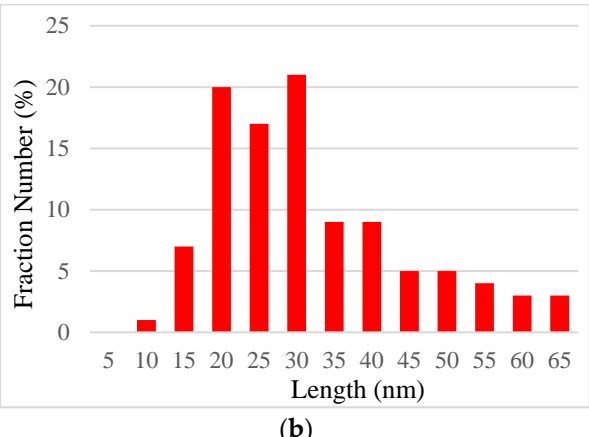

(**a**)                                        (**b**)

**Figure 9.** Variation of the length of η-Ni$_3$Ti rod-like particles in lath martensite after (**a**) quality heat treatment, and after (**b**) ageing at 475 °C for 3000 h.

EDX microanalysis revealed that η-Ni$_3$Ti particles after long-term ageing at 475 °C were enriched with molybdenum, chromium, and iron (Table 6). Diffraction contrast in martensite around η-Ni$_3$Ti particles proved the presence of elastic stresses induced by precipitation. The number density of η-Ni$_3$Ti particles in the martensite matrix was variable. It indicates preferred nucleation around some primary precipitates that interact with strain fields of the others [20]. Uneven density of precipitates gives rise to nonhomogeneous mechanical properties on the microscopic scale. No recrystallization of lath martensite was observed. Even after exposure for 3000 h at 475 °C, martensitic laths were preserved.

**Table 6.** Results of semiquantitative EDX analyses of η-Ni$_3$Ti particles after ageing at 475 °C for 3000 h (at. %).

| No. | Mo | Ti | Cr | Fe | Ni |
|-----|-----|------|-----|-----|------|
| 1 | 6.4 | 24.7 | 5.1 | 1.3 | 62.6 |
| 2 | 5.7 | 23.4 | 4.2 | 1.4 | 65.3 |
| 3 | 6.2 | 23.6 | 2.2 | 1.1 | 66.8 |
| Average | 6.1 | 23.9 | 3.8 | 1.3 | 64.9 |

Detailed studies of precipitation processes using carbon-extraction replicas revealed that annealing at 475 °C was accompanied by the formation of Cr-rich α'nanometric particles. These particles are marked in Figure 8b. The typical size of Cr-rich α'precipitates was only several nanometers. Precipitation of Cr-rich α'particles in the Fe–Cr alloys is related to the decomposition of the matrix to a mixture of α' + α phases in the temperature interval of approximately 425–550 °C [40]. Results of semiquantitative EDX analyses of the Cr-rich α'particles are stated in Table 7. As evident, these particles also contain about 15 at.% of molybdenum, which is, similarly to chromium, a strong ferrite stabilizing element [3]. Molybdenum enrichment of Cr-rich α'particles was also reported in [12]. It is known that this solid solution decomposition is accompanied by increasing of hardness and strength of Fe–Cr alloys. However, the chromium content in the cast investigated was only 11.07 wt.%, and the fraction of nanometric Cr-rich α'particles was too low to cause a significant strengthening effect.

**Table 7.** Results of semiquantitative EDX analyses of Cr-rich $\alpha'$ phase after ageing at 475 °C for 3000 h (at.%).

| No. | Cr | Fe | Mo |
|---|---|---|---|
| 1 | 72.2 | 12.2 | 15.6 |
| 2 | 69 | 16.2 | 14.8 |
| 3 | 72.3 | 11.6 | 17.1 |
| Average | 70.8 | 13.3 | 15.8 |

Figure 10a shows the substructure of a martensite lath in the sample after annealing at 475 °C for 3000 h containing intensive precipitation of several variants of $\eta$-Ni$_3$Ti particles. Figure 10b shows a composed spot diffraction pattern from this area in the zone axis [110]$\alpha$. There are weak extra spots which do not belong to the matrix. Extra spots in this diffraction pattern can be indexed as two variants of a hexagonal $\omega$ phase ($a_\omega$ = 0.404 nm, $c_\omega$ = 0.248 nm) or two variants of {112} <111> twinning in the BCC matrix (Figure 10c–e). The interpretation as $\omega$ phase is not rational for the following reasons:

— Extensive TEM studies of precipitation reactions in lath martensite using carbon extraction replicas revealed only precipitation of $\eta$-Ni$_3$Ti and Cr-rich $\alpha'$ phases. No Mo-enriched particles of metastable $\omega$ phase [12] were detected.
— Between the BCC and hexagonal phases usually exists the Burgers orientation relationship [15]. Nevertheless, zone axes of hexagonal $\omega$ phase in Figure 10c do not obey the Burgers orientation relationship with the martensitic matrix.

The indexation of extra spots in the composed diffraction pattern in Figure 10d,e as two variants of {112} <111> twinning and intensive double diffraction effects on overlapping interfaces explains all extra spots observed in Figure 10b.

Chen et al. [9] reported that some martensitic laths in CUSTOM 465® steel after cryogenic treatment as well as after subsequent ageing at 480–640 °C contained thin parallel twins. They interpreted this "interpenetrating twin structure" as a consequence of martensitic transformation—in the case of the Kurdjumov–Sachs orientation relationship between parent austenite and martensite, some martensite variants obey the twin-related orientation [3]. Twinning in low-carbon lath martensite was also studied in [13,30]. Zhang et al. [30] explained double reflection spots due to overlapping of twins and the matrix as nanoscale $\omega$ precipitates. Published diffraction patterns related to twinning in low-carbon lath martensite contained just one variant of {112} <111> twinning, as expected in the case of transformation twins in martensite [9,13,30]. However, the composite diffraction pattern from a single martensite lath in Figure 10b comprises two variants of twinning. Extra spots in Figure 10b are weak and exhibit faint diffusional streaking along <112>$^*_\alpha$ directions. This means that twins are very thin. They grow along the same directions in the BCC matrix as $\eta$-Ni$_3$Ti rods, i.e., along <111>$_\alpha$. Chen et al. [9] reported that EBSD investigations revealed significant residual stresses in the aged martensite of CUSTOM 465® stainless steel. These residual stresses around several variants of $\eta$-Ni$_3$Ti rods could have enabled the formation of several variants of very thin twins inside martensitic laths.

The fraction of reverted austenite in the sample after ageing at 475 °C for 3000 h was determined by XRD analysis as 11.5 $\pm$ 0.5 wt.%. Metallographic analysis revealed bright areas in aged martensite, mostly in segregation bands. These blocky particles of irregular shape with size of up to several micrometers were proved using EBSD to be reverted austenite. Figure 11a,b show these reverted austenite particles on an LM micrograph and on an EBSD phase map, respectively. The number frequency of blocky particles of reverted austenite and their size in segregation bands were much greater than in the surrounding matrix. The average nickel content in segregation bands, as determined by EDX microanalysis, was 12.5 wt.%, while in the surrounding matrix it was only 9.7 wt.%. This proves the importance of nickel in chemical stabilization of reverted austenite in aged martensite of maraging steels.

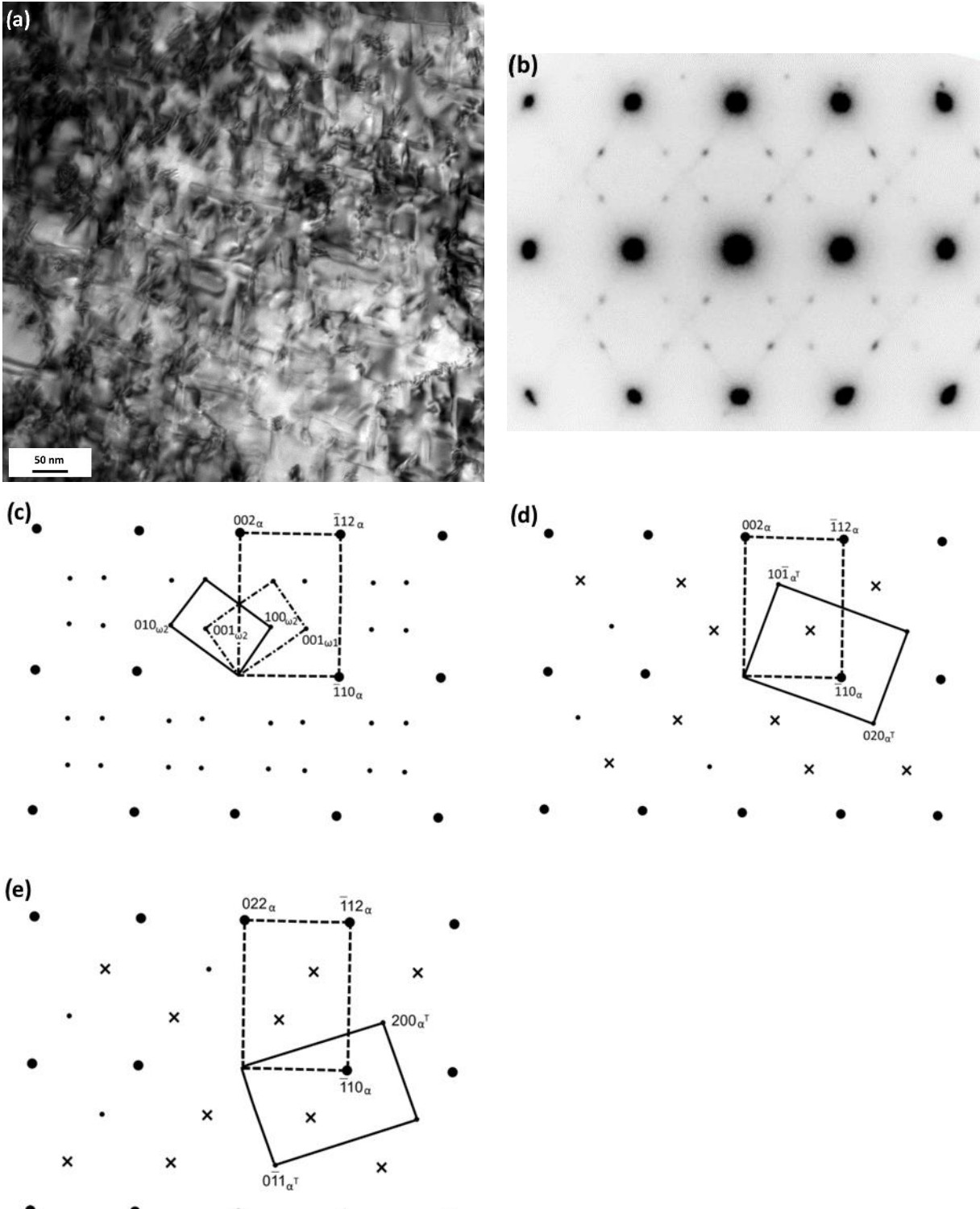

**Figure 10.** Diffraction analysis of lath martensite after ageing at 475 °C for 3000 h. (**a**) Intensive precipitation in martensite, bright field image; (**b**) the composed spot diffraction pattern; (**c**) zone axes: [110]α + [100]ω1 + [010]ω2; (**d**) zone axes: [110]α + [101]αT; (**e**) zone axes: [110]α + [011]αT. α—matrix, αT—twins, x—double diffraction spots.

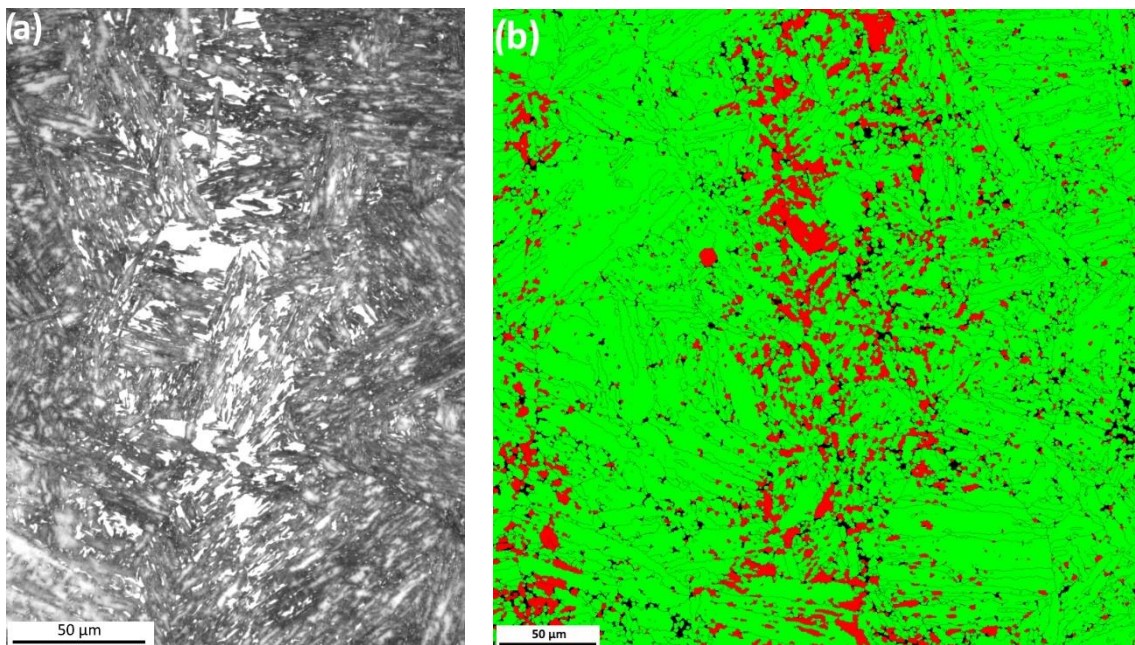

**Figure 11.** Reverted austenite in segregation bands in martensite after ageing at 475 °C for 3000 h. (**a**) Bright particles of reverted austenite in aged martensite, LM. (**b**) EBSD phase map; green = martensite, red = reverted austenite.

TEM analysis revealed that reverted austenite was also present in the forms of thin films between martensite laths and small intralath particles (Figure 12a–c). Dislocation density in reverted austenite was low and no precipitates were present in this phase. These morphologies and size of reverted austenite are assumed to be very beneficial for high ductility and toughness of maraging steels.

Inconspicuous changes in mechanical properties of CUSTOM 465® stainless steel during long-term ageing at 475 °C are closely related to microstructural evolution during ageing. Ageing was accompanied by gradual recovery of martensite and precipitation processes. No recrystallization of martensite was observed. The main contribution to precipitation strengthening of low-carbon martensite was the intensive precipitation of rod-like η-Ni$_3$Ti particles [9]. The growth of these particles during long-term aging at 475 °C was relatively slow. Softening of martensite due to growth of η-Ni$_3$Ti particles was partly compensated by additional precipitation of η-Ni$_3$Ti particles induced by solubility differences at 524 and 475 °C and by coprecipitation of nanometric Cr-rich α′particles. Due to the relatively low chromium content in CUSTOM 465® stainless steel, the number density of Cr-rich particles was much lower than that of η-Ni$_3$Ti particles. The decomposition of the BCC solid solution affected precipitation hardening much less than the intensive precipitation of the η-Ni$_3$Ti phase. Precipitation processes in lath martensite were accompanied by the formation of reverted austenite. The fraction of reverted austenite in the microstructure after annealing at 475 °C for 3000 h was nearly double that after quality heat treatment. Reverted austenite contributed to softening of martensite but also positively affected ductility and toughness of the steel investigated [26].

APT results on precipitation reactions in the Cr–Ni–Ti–Mo maraging steels indicated the importance of synergistic alloying effects on nanoscale precipitation [11,12]. Detailed investigations on precipitation reactions in a Cr–Ni–1Ti–3Mo alloy revealed that partitioning of molybdenum in η-Ni$_3$Ti particles during ageing enabled the formation of nanometric Mo-enriched particles, which were identified by diffraction analysis as ω phase [12]. Detailed TEM investigations on precipitation reactions in CUSTOM 465 stainless steel after long-term aging at 475 °C did not reveal the formation of ω phase. Diffraction effects which are sometimes attributed to metastable ω phase [29,30] have been interpreted as

{112}<111>twinning in martensite [3]. It is worth noting that two variants of twinning were observed in a single martensitic lath. It is likely that these thin twins did not form during martensitic transformation but only during subsequent ageing.

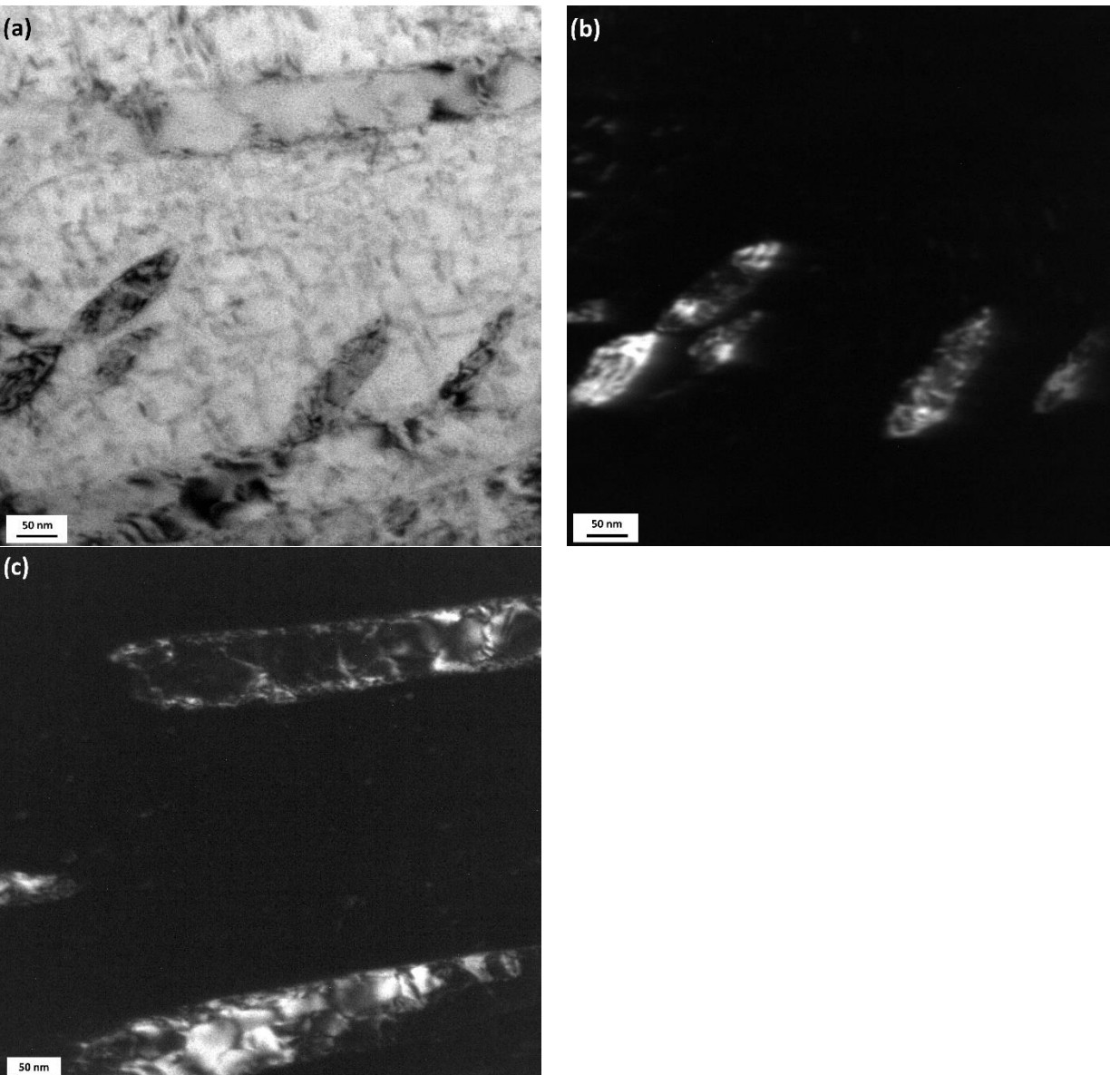

**Figure 12.** Interlath films and intralath particles of reverted austenite in lath martensite after ageing at 475 °C for 3000 h. (**a**) Bright field image; (**b**) dark field image of reverted austenite in $200_\gamma$ reflection; (**c**) dark field image of reverted austenite in $111_\gamma$ reflection.

Changes in strength and hardness in CUSTOM 465® stainless steel during long-term ageing at 475 °C were affected by gradual recovery processes in lath martensite, precipitation strengthening of martensite, and the fraction of reverted austenite particles. It can be assumed that reverted austenite positively affected ductility during tensile testing due to the transformation-induced plasticity (TRIP) effect [26,27].

## 4. Conclusions

Results of microstructural characterization and mechanical property evaluation in CUSTOM 465® stainless steel after long-term ageing at 475 °C can be summed up as follows:

1.  Ageing of the steel at 475 °C for 1000, 2000, and 3000 h was accompanied by a gradual slight decline in strength. The longest exposure led to a drop in yield strength of only 8% and 10% in the case of ultimate tensile strength. On the other hand, ductility and toughness values remained almost unchanged.

2.  During ageing at 475 °C, additional precipitation of nanometric η-Ni$_3$Ti particles in lath martensite took place. This is the most important minor phase in the alloy investigated. Furthermore, ageing at 475 °C resulted in decomposition of the BCC solid solution ($\alpha$), which was accompanied by the formation of Cr-rich particles of $\alpha'$ phase. The formation of this phase partly compensated the decline of strength due to slow growth of η-Ni$_3$Ti particles and the formation of reverted austenite.

3.  Diffraction studies revealed the existence of {112} <111> twinning in lath martensite. Two variants of thin twins were found in a single martensitic lath. It is likely that these thin twins did not form during martensitic transformation but only during subsequent ageing.

4.  The fraction of reverted austenite after ageing for 3000 h was approximately double compared to that in the state after quality heat treatment. Reverted austenite formed blocks, interlath films, and small intralath particles. Particles of reverted austenite were free of precipitates.

5.  Slow kinetics of martensite recovery and growth of η-Ni$_3$Ti rods, and additional precipitation of η-Ni$_3$Ti and Cr-rich particles of $\alpha'$ phase minimized the decrease of martensite strength during the long-term ageing at 475 °C. At the same time, the formation of reverted austenite contributed to softening of martensite, but also had a positive effect on ductility and toughness of the alloy investigated. No susceptibility to 475 °C embrittlement was proved.

**Author Contributions:** V.V.: conceptualization, TEM investigations, data curation, writing—original draft. G.R.: SEM investigations, data curation, visualization. Z.K.: investigations—heat treatment and mechanical properties, supervision, writing—review and editing, funding acquisition. A.V.: EBSD investigations, visualization. R.P.: writing—original draft, validation, visualization. All authors have read and agreed to the published version of the manuscript.

**Funding:** This research was funded by the European Union and the state budget of the Czech Republic, the project no. CZ.02.1.01/0.0/0.0/17_048/0007373.

**Data Availability Statement:** Not applicable.

**Acknowledgments:** This contribution was created with financial support from the project no. CZ.02.1.01/0.0/0.0/17_048/0007373 "Damage Prediction of Structural Materials" within the Research, Development and Education Operational Programme financed by the European Union and from the state budget of the Czech Republic and the project no. SP2022/33 "Study of the relationship between the microstructure and properties of progressive technical materials, degradation mechanisms and behavior of progressive technical materials in different operating conditions". R.P. gives thanks for financial support from the project no. 00424/2022/RRC funded by the Moravian-Silesian Region "Support for gifted students of doctoral studies at VŠB-TUO". This paper was created as part of the drawing and use of institutional support for long-term and conceptual development of a research organization in 2022, provided by the Ministry of Industry and Trade of the Czech Republic.

**Conflicts of Interest:** The authors declare no conflict of interest.

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
