# Peer review of "The Effect of Long-Term Ageing at 475 °C on Microstructure and Properties of a Precipitation Hardening MartensiticStainless Steel"

_metals, doi:10.3390/met12101643_

Round 1

Reviewer 1 Report

This paper investigated the effect of long-term aging at 475 ℃ on microstructure and properties of CUSTOM 465 stainless steel. Light microscopy (LM), scanning electron microscopy (SEM), transmission electron microscopy (TEM) and X-ray diffractometry (XRD) were used to characterize the microstructural evolution, especially the nanometric particles. Tensile tests and impact tests were carried out study the mechanical properties. This manuscript can be accepted for publication if the following points are taken into consideration.

1.     Page 1, should be “{112} <111>twins” and “Precipitation of ω phase”.

2.     The number of keywords is too many.

3.     Page 3 paragraph 3, this discussion about copper is redundant since the results of this paper do not involve Cu-rich precipitates.

4.     “Section 2. Experimental material and procedures” should be divided into “Section 2. Experimental material and procedures” and “Section3. Results”.  This point is mandatory prior to publication.

5.     The detailed XRD parameters and calculation method of volume fraction of austenite based on XRD data are missing.

Author Response

Answers to the Reviewer 1

This paper investigated the effect of long-term aging at 475 ℃ on microstructure and properties of CUSTOM 465 stainless steel. Light microscopy (LM), scanning electron microscopy (SEM), transmission electron microscopy (TEM) and X-ray diffractometry (XRD) were used to characterize the microstructural evolution, especially the nanometric particles. Tensile tests and impact tests were carried out study the mechanical properties. This manuscript can be accepted for publication if the following points are taken into consideration.

  1. Page 1, should be “{112} <111>twins” and “Precipitation of ω phase”.

It is probably related to the Abstract. In the original manuscript it was OK. However, we were asked to shorten the Abstract. In the Revised manuscript the sentences containing these terms were omitted.

  1. The number of keywords is too many.

The number of keywords was reduced.

  1. Page 3 paragraph 3, this discussion about copper is redundant since the results of this paper do not involve Cu-rich precipitates.

Our steel contained some copper. That is why Cu-rich particles were discussed, as they can be important for nucleation of reverted austenite. It is true that no Cu-rich particles were identified during experimental studies. Nevertheless, for a complex description of possible precipitation reactions in maraging steels we would be glad if this short paragraph could be preserved. Thank you.

  1. “Section 2. Experimental material and procedures” should be divided into “Section 2. Experimental material and procedures” and “Section3. Results”.  This point is mandatory prior to publication.

Yes, it was done.

  1. The detailed XRD parameters and calculation method of volume fraction of austenite based on XRD data are missing.

Information about XRD methodology was added.

Reviewer 2 Report

The manuscript titled "The effect of long-term ageing at 475 °C on microstructure and properties of CUSTOM®465 stainless steel" (authors: Vlastimil Vodárek, Gabriela Rožnovská, Zdeněk Kuboň, Anastasia Volodarskaja and Renáta Palupčíková) corresponds to the scope of "Metals".

The manuscript has quite many comments, thus it requires minor revisions to be made.

1. The commercial steel grade CUSTOM®465 should not be given in the title of the manuscript. It is recommended to use a steel type designation that is more understandable to the readers, such as Ni-Cr-Ti-Mo maraging steels.

2. In the methodology. It is worth adding information for SEM, in particular the accelerating voltage, for XRD - the type of filter, X-ray wavelength, scanning angles and what base was used to determine the phases in the composition of the alloy. It remains unclear what the samples used to determine the hardness were (flat, cylindrical), how the surface was prepared for the analysis. Usually, after hardening and other strengthening treatment, the hardness is determined by the Rockwell method (scale "C"). Impact tests were apparently carried out according to some kind of standard, similar to the test for stretching. It is worth adding the title and number of the standard or removing this information everywhere. How was the cutting of test samples from a massive block carried out?

3. It is difficult to compare the microstructures in Figure 1, as they were obtained at different magnifications. It is necessary to use the same conditions for image production or justify the choice of a different magnification (scale).

4. Figure 3 shows data on the presence of reverted austenite (RA). However, there is no information on the presence of other TiX (X = N,C) compounds, which were mentioned earlier in the text. It is worth showing with arrows which structures are observed in the image, e.g. rod-like Ni3Ti.

5. What does the symbol "Z", which has the dimension MPa, mean in tables 3 and 4? In table 4, the dimension corresponds to %. It is necessary to choose the right option, most likely the last one.

6. In Figs. 4,5 there are points, apparently the arithmetic mean. However, there is no indication of a confidence interval. The graph of approximation of experimental values ​​should be represented by curves or splines, but not by segments.

7. Fig. 6 is also made at different magnifications. It is not clear what distinguishes these types of structure from the results of topography (surface morphology) analysis of fractograms.

8. It is worth combining Fig.3 and Fig.7. This will help to more clearly see the difference in the microstructure after ageing.

9. The Discussion section is not very large, so it can be combined with the results part. However, this decision is up to the authors.

Author Response

Answers to the Reviewer 2

The manuscript has quite many comments, thus it requires minor revisions to be made.

  1. The commercial steel grade CUSTOM®465 should not be given in the title of the manuscript. It is recommended to use a steel type designation that is more understandable to the readers, such as Ni-Cr-Ti-Mo maraging steels.

The effect of long-term ageing at 475 °C on microstructure and properties of a Cr – Ni – Ti - Mo maraging steel

  1. In the methodology. It is worth adding information for SEM, in particular the accelerating voltage, for XRD - the type of filter, X-ray wavelength, scanning angles and what base was used to determine the phases in the composition of the alloy. It remains unclear what the samples used to determine the hardness were (flat, cylindrical), how the surface was prepared for the analysis. Usually, after hardening and other strengthening treatment, the hardness is determined by the Rockwell method (scale "C"). Impact tests were apparently carried out according to some kind of standard, similar to the test for stretching. It is worth adding the title and number of the standard or removing this information everywhere. How was the cutting of test samples from a massive block carried out?

“Quality heat treatment and long-term ageing were carried out in an electric furnace with a protective atmosphere. Transformation temperatures during heating and cooling of the alloy were determined using the quenching dilatometer Bähr DIL 805A. The heating rate in the interval of phase transformations was 1 °C/min. The cooling rate from the temperature of 950 °C was 3 °C/min. up to a temperature of 35 °C. Temperature of the end of martensitic transformation was lower than the temperature at the end of the measurement and therefore could not be determined. Microstructure and basic mechanical properties were studied in the state after quality heat treatment and after additional ageing at 475 °C for 1000, 2000 and 3000 hours. Microstructural characterization was carried out using light microscopy (LM), scanning electron microscopy (SEM), transmission electron microscopy (TEM) and X-ray diffractometry (XRD). XRD measurements were carried out using a Bruker – AXS D8 Advance diffractometer equipped with a position sensitive detector LynxEye. The following parameters were applied: radiation CuKa, absorption filter Ni, voltage 40 kV, current 40 mA, step 0.014° in the interval of 2q angles 40 – 135° and digital processing of data using software Bruker Diffract Suite. Diffraction database PDF-2 was applied for qualitative analysis, quantification was based on the Rietveld´s method using the software Bruker Topas, version 4.2.

Metallographic samples were oriented perpendicular and parallel to the rod´s axis and were cut in approximately one-quarter the diameter of the rod. Polished samples were etched in a V2A solution (a mixture of 10 ml HNO3, 100 ml HCl and 100 ml water) and observed in an Olympus GX51 microscope. SEM was performed using a Quanta 450FEG microscope equipped by X-ray microanalysis (EDX) and electron backscattered diffraction (EBSD) facilities. Topography of fracture surfaces was recorded using a signal in secondary electrons (SE). EBSD mapping was carried out on specimens after polishing on colloidal silica at accelerating voltage of 15 kV and a step size of 0.2 mm was applied. TEM analysis was performed on a JEM 2100 microscope equipped with an EDX analyser. Identification of phases was carried out by a combination of EDX microanalysis and electron diffraction. Both carbon extraction replicas and thin foils were used. Thin foils were prepared by twin-jet electrolytic polishing to perforation using a Struers Tenupol 2 equipment. Electropolishing was carried out in an electrolyte consisting of 5 vol.% perchloric acid, 20 vol.% glycerol and 75 vol.% ethanol at – 5 °C and a potential of 20 V. Tensile tests were carried out at room temperature according to the ASTM E8 standard on an MTS 810 testing machine at a strain rate of 1 x 10-3 s-1. Tensile bars of do = 6 mm and l = 5do were oriented in the tangential direction of the original rod. Impact tests were carried out on samples oriented in the same way, with a V – notch and dimensions of 5 x 10 x 55 mm. Tests were performed at room temperature on a Charpy instrument 300 J according to the ASTM E23 standard. Hardness measurements were carried out on metallographic specimens using the Vickers method at a load of 30 kg.”   

  1. It is difficult to compare the microstructures in Figure 1, as they were obtained at different magnifications. It is necessary to use the same conditions for image production or justify the choice of a different magnification (scale).

Micrographs at the same magnification have been used and a detail has been added at higher magnification.

  1. Figure 3 shows data on the presence of reverted austenite (RA). However, there is no information on the presence of other TiX (X = N,C) compounds, which were mentioned earlier in the text. It is worth showing with arrows which structures are observed in the image, e.g. rod-like Ni3Ti.

As mentioned in the text, the typical size of TiX particles was several micrometres. They cannot be present in Fig. 3. As described in the caption, most particles are Ni3Ti phase. TiX particles are documented in Figs. 1b,c and Figs. 6a,b.

  1. What does the symbol "Z", which has the dimension MPa, mean in tables 3 and 4? In table 4, the dimension corresponds to %. It is necessary to choose the right option, most likely the last one.

Yes, you are right, units for Z were wrong. Furthermore, we replaced the letter A by the international symbol L (elongation) and the letter Z by the international shortcut R. A. (reduction of area).

Table 3 Mechanical properties after quality heat treatment

Rp0.2

[MPa]

Rm

 [MPa]

L

 [%]

R.A.

[%]

HV 30

KV

 [J]

1531

1617

11.3

44.0

500

18

  1. In Figs. 4,5 there are points, apparently the arithmetic mean. However, there is no indication of a confidence interval. The graph of approximation of experimental values ​​should be represented by curves or splines, but not by segments.

As described in the manuscript, points in Figs. 4 and 5 represent the arithmetic means of data presented in Table 4. There are no confidence intervals, because arithmetic means for each point are based on results of two tensile tests. The style of Figs. 4 and 5 has been modified to curves.

  1. Fig. 6 is also made at different magnifications. It is not clear what distinguishes these types of structure from the results of topography (surface morphology) analysis of fractograms.

Micrographs at the same magnification have been used and captions have been modified.

  1. It is worth combining Fig.3 and Fig.7. This will help to more clearly see the difference in the microstructure after ageing.

Figs. 3 and 7 are parts of the text paragraphs that comprehensively describe microstructure of samples with different thermal history. Therefore, we would like to present them separately.   

  1. The Discussion section is not very large, so it can be combined with the results part. However, this decision is up to the authors.

Both chapters have been merged.

Reviewer 3 Report

Review report: The effect of long-term ageing at 475 °C on microstructure and properties of CUSTOM 465 stainless steel

1.       Shorten the length of the abstract section and add only key information.

2.       Discuss about the selection of the ageing temperature.

3.       Application and novelty of the work should be in a separate section.

4.       Shorten the length of the introduction section and add key published work and try to make a bridge between current and previous published work. Refer to some recently published work on heat treatment: https://doi.org/10.1007/s12666-015-0826-z.

5.       The purpose of the initial heat treatment is also not discussed.

6.       Add the reference for Table 2.

7.       In Fig. 1, lath martensite is clear but not inclusions. Add good quality images with higher magnification.

8.       Add the complete detail of the tensile testing, specimen, fractured specimen image, stress-strain curve and tensile results in a separate table.

9.       Fractured specimen is presented without any discussion add the complete detail about tear ridges, cleavage, dimples and also the mechanism of the failure: https://doi.org/10.1016/j.engfailanal.2016.06.012.

10.    Improve the technical content and add more references in discussion section.

Author Response

Answers to the Reviewer 3

  1. Shorten the length of the abstract section and add only key information.

The Abstract has been shortened:

“The effect of long-term ageing (1000, 2000 and 3000 hours) at 475°C on mechanical properties, microstructure and substructure of CUSTOM 465â maraging stainless steel was studied. The additional precipitation of nanometric particles of h-Ni3Ti phase in partly recovered lath martensite and decomposition of the BCC solid solution accompanied by the formation of nanometric Cr-rich α´particles were identified. The fraction of reverted austenite in the final microstructure gradually increased with time of ageing at 475 °C. Ageing resulted in a gradual slight decline (up to 10 %) in yield strength, ultimate tensile strength and hardness. On the other hand, for all ageing dwells ductility and impact energy values remained almost unchanged. The reason for this phenomenon lies in the gradual increase in the fraction of reverted austenite during long-term ageing at 475 °C and at the same time in the sluggish kinetics of microstructural changes in lath martensite. No susceptibility to 475 °C embrittlement has been proved.”

  1. Discuss about the selection of the ageing temperature.

We believe that the selection of ageing temperature is clear from the 7th paragraph of Introduction and from the goal of the work:

“Long-term thermal exposure of Fe-Cr based alloys containing from 12 to 70 wt.%Cr between 425 and 550 °C may significantly deteriorate properties of stainless steels [37]. This 475 °C embrittlement manifests itself by increased hardness and ductile-brittle transition temperature [38]. The reason for this phenomenon is the existence of the miscibility gap in the binary Fe-Cr phase diagram [37].”

and from the paragraph 8th:

“The published data [4] show the effect of thermal exposure in the temperature interval of 316 – 482 °C for 1000 hours on mechanical properties of the alloy. Long-term exposure at 371 and 427 °C was accompanied by a slight strengthening and a gradual decrease in impact toughness. On the other hand, long-term exposure at temperature of 482 °C indicated a decline in strength values while impact energy increased [4]. CUSTOM 465â stainless steel has the chromium content close to the lower limit which is regarded to be critical from the point of view of the Fe-Cr solid solution decomposition and developing of 475 °C embrittlement during long-term exposure at temperatures in the range of about 425 to 550 °C.”

  1. Application and novelty of the work should be in a separate section.

The goal of the work is defined in the last paragraph of Introduction:

“The main objective of this paper is to investigate the effects of exposure at 475 °C for 1000, 2000 and 3000 hours on the microstructural evolution and mechanical properties of CUSTOM 465â stainless steel. Results are expected to deepen the knowledge about long-term microstructural stability, precipitation reactions, mechanical properties of Ni-Cr-Ti-Mo maraging steels and their susceptibility to 475 °C embrittlement.”

  1. Shorten the length of the introduction section and add key published work and try to make a bridge between current and previous published work. Refer to some recently published work on heat treatment:https://doi.org/10.1007/s12666-015-0826-z.

The introduction summarizes current information on precipitation reactions in Cr-Ni-Ti-Mo maraging steels. Precipitation plays a key role in strengthening of these steels. In the Results section, this comprehensive information is used when discussing our results with current knowledge. Therefore, we think that the length of the Introduction section is justified.

The recommended reference about heat treatment is dealing with a different type of steels: heat resistant P91 steel. Detailed information about heat treatment of maraging steels is presented in references [1-24].

  1. The purpose of the initial heat treatment is also not discussed.

Generally, the goal of the quality heat treatment is to obtain the optimal structural state and properties of the alloy.  In the Introduction section it is mentioned that the peak-aged condition of Custom 465Ò steel can be achieved after ageing at about 520 °C. This ageing temperature was used at the design of the quality heat treatment regime – a sentence has been added to the text:

“The ageing parameters were designed to be close to the peak-aged condition of the steel investigated [9].” 

  1. Add the reference for Table 2.

Table 2 contains experimental results of our dilatometric investigations. No reference is needed.

  1. In Fig. 1, lath martensite is clear but not inclusions. Add good quality images with higher magnification.

Yes, a micrograph at higher magnification has been added.

  1. Add the complete detail of the tensile testing, specimen, fractured specimen image, stress-strain curve and tensile results in a separate table.

Details of tensile testing are in the chapter Experimental material and procedures. Tensile results are summarized in Table 4.

Results of fractographic analysis on broken tensile specimens have been added.

  1. Fractured specimen is presented without any discussion add the complete detail about tear ridges, cleavage, dimples and also the mechanism of the failure: https://doi.org/10.1016/j.engfailanal.2016.06.012.

The interpretation of topography of fracture surfaces in Figs. 7a,b is given in the paragraph just above the micrographs.

      Additional information has been added to captions of Figs. 7a,b. The reference https://doi.org/10.1016/j.engfailanal.2016.06.012 has been added.

  1. Improve the technical content and add more references in discussion section.

The technical content of the paper has been improved. The Discussion section has been merged with the Results and some references have been added. 

Round 2

Reviewer 3 Report

Accept.

Author Response

Shorten the length of the introduction section and add key published work and try to make a bridge between current and previous published work. Refer to some recently published work on heat treatment: https://doi.org/10.1007/s12666-015-0826-z.

The introduction summarizes current information on precipitation reactions in Cr-Ni-Ti-Mo maraging steels. Precipitation plays a key role in strengthening of these steels. In the Results section, this comprehensive information has been used when discussing our results with current knowledge. Therefore, we do think that the length of the Introduction section is justified.

To our best knowledge all significant  papers dealing with Cr-Ni-Ti-Mo maraging steels published up to 2022 have been properly cited. Detailed information about heat treatment of maraging steels is presented in references [1-24].